# Micro-structure diffusion scalar measures from reduced MRI acquisitions

**Santiago Aja-Fernández**[1,3]*, **Rodrigo de Luis-García**[1], **Maryam Afzali**[3],
**Malwina Molendowska**[3], **Tomasz Pieciak**[2], **Antonio Tristán-Vega**[1]

**1** Laboratorio de Procesado de Imagen (LPI), Universidad de Valladolid, Spain, **2** AGH University of Science and Technology, Krakow, Poland, **3** Cardiff University Brain Research Imaging Center (CUBRIC), School of Psychology, University of Cardiff, UK

* sanaja@tel.uva.es

**Data Availability Statement:** All the experiments reported in the manuscript have been undergone using publicly available data sets, both for control subjects and for diseased subjects, as described

## Abstract

In diffusion MRI, the Ensemble Average diffusion Propagator (EAP) provides relevant micro-structural information and meaningful descriptive maps of the white matter previously obscured by traditional techniques like Diffusion Tensor Imaging (DTI). The direct estimation of the EAP, however, requires a dense sampling of the Cartesian q-space involving a huge amount of samples (diffusion gradients) for proper reconstruction. A collection of more efficient techniques have been proposed in the last decade based on parametric representations of the EAP, but they still imply acquiring a large number of diffusion gradients with different b-values (shells). Paradoxically, this has come together with an effort to find scalar measures gathering all the q-space micro-structural information probed in one single index or set of indices. Among them, the return-to-origin (RTOP), return-to-plane (RTPP), and return-to-axis (RTAP) probabilities have rapidly gained popularity.

In this work, we propose the so-called "Apparent Measures Using Reduced Acquisitions" (AMURA) aimed at computing scalar indices that can mimic the sensitivity of state of the art EAP-based measures to micro-structural changes. AMURA drastically reduces both the number of samples needed and the computational complexity of the estimation of diffusion properties by assuming the diffusion anisotropy is roughly independent from the radial direction. This simplification allows us to compute closed-form expressions from single-shell information, so that AMURA remains compatible with standard acquisition protocols commonly used even in clinical practice. Additionally, the analytical form of AMURA-based measures, as opposed to the iterative, non-linear reconstruction ubiquitous to full EAP techniques, turns the newly introduced *apparent* RTOP, RTPP, and RTAP both robust and efficient to compute.

## Introduction

Under the name of Diffusion Magnetic Resonance Imaging (DMRI) we gather a set of diverse MRI imaging techniques with the ability of extracting *in vivo* relevant information regarding the random, anisotropic diffusion of water molecules that underlay the structured nature of

therein. In particular, two databases were used: - The Human Connectome Project (HCP), (https://ida.loni.usc.edu/login.jsp). The HCP project (Principal Investigators: Bruce Rosen, M.D., Ph.D., Martinos Center at Massachusetts General Hospital; Arthur W. Toga, Ph.D., University of Southern California, Van J. Weeden, MD, Martinos Center at Massachusetts General Hospital) is supported by the National Institute of Dental and Craniofacial Research (NIDCR), the National Institute of Mental Health (NIMH), and the National Institute of Neurological Disorders and Stroke (NINDS). HCP is the result of efforts of co-investigators from the University of Southern California, Martinos Center for Biomedical Imaging at Massachusetts General Hospital (MGH), Washington University, and the University of Minnesota. - The Public Parkinson's Disease database (PPD), acquired at the Cyclotron Research Centre, University of Liège. Available: https://www.nitrc.org/frs/?group_id=835.

**Funding:** S. Aja-Fernández: supported by Ministerio de Ciencia e Innovación of Spain with research 581 grants RTI2018-094569-B-I00 and PRX18/00253 (Estancias de profesores e 582 investigadores senior en centros extranjeros). The funders had no role in study design, data collection and analysis, decision to publish, or preparation of the manuscript. M. Afzali: supported by a 583 Wellcome Trust grant (096646/Z/11/Z). The funders had no role in study design, data collection and analysis, decision to publish, or preparation of the manuscript. T. Pieciak: supported by National Science 584 Centre (Poland); funding resource (2015/19/N/ST7/01204). The funders had no role in study design, data collection and analysis, decision to publish, or preparation of the manuscript. S. Aja-Fernández; R. de Luis-García; A. Tristán-Vega; supported by grant number 18ISZD/541A302/692 of the Ministerio de Educación, Cultura y Deporte. Government of Spain. The funders had no role in study design, data collection and analysis, decision to publish, or preparation of the manuscript.

**Competing interests:** The authors have declared that no competing interests exist.

different living tissues. It has attracted an extraordinary interest among the scientific community over recent years due to the relationships found between a number of neurological and neurosurgical pathologies and alterations in the white matter as revealed by an increasing number of DMRI studies [1–3].

The most relevant feature of DMRI is its ability to measure directional variance, i.e. anisotropy. In the beginning of the 2000s, diffusion tensor MRI [4, DT-MRI] gained huge popularity in white matter studies, not only among technical researchers but also among clinical partners, to the point that even nowadays most of the research studies involving DMRI focus on the diffusion tensor (DT). By using a simple Gaussian regressor, the anisotropy of the tissues is actually probed by acquiring as few as 20 to 60 images, which is acceptable in clinical practice. DT-MRI brought to light one of the most common problems in DMRI techniques: in order to carry out clinical studies, the information given by the selected diffusion analysis method must be translated into some scalar measures that describe different features of the diffusion within every voxel. That way, measures like the Fractional Anisotropy (FA), the Axial and Radial Diffusivity (AD, RD) or the Mean Diffusivity (MD) were defined [5]. Even at the early stages of DT-MRI, it was clear that the Gaussian assumption had important limitations. It provided a useful tool allowing clinical studies, but the underlying diffusion processes were not accurately described because of the over-simplified fitting, so that more evolved techniques with more degrees-of-freedom naturally arose, such as High Angular Resolution Diffusion Imaging [6–8, HARDI] or Diffusion Kurtosis Imaging [9, DKI]. It seems obvious that more degrees-of-freedom require more diffusion images to be acquired, but the requirement of an accurate angular resolution also implies the need for a finer angular contrast, which translates in the need for stronger gradients to probe diffusion, i.e., higher b-values [10].

The trend over the last decade has consisted in acquiring a large number of diffusion-weighted images distributed over several shells together with moderate-to-high b-values to estimate more advanced diffusion descriptors, such as the Ensemble Average diffusion Propagator [11, 12, EAP]. This leads to a completely model-free, non parametric approach for diffusion that can accurately describe most of the relevant phenomena associated to diffusion.

The most straightforward way of estimating the EAP is Diffusion Spectrum Imaging [11, DSI], that relies on the dense sampling of the q-space for discrete Fourier transformation. Hence, it requires a huge number of images to avoid aliasing artifacts and attain a decent accuracy, which makes it not so appealing in practice. As a consequence, alternative techniques aim to parametrically reconstruct the EAP from reduced samplings of the q-space, most of them related to the recent advances in compressed sensing and sparse reconstruction [13, 14]. In practice, some multi-shell reconstruction techniques may be used to compute the EAP, typically as a superposition of the integrals analytically computed for each basis function. Some of the most prominent methods are *Hybrid Diffusion Imaging* [15, HYDI], the *multiple q-shell diffusion propagator imaging* [16, 17, mq-DPI], the *Bessel Fourier Orientation Reconstruction* [18, BFOR], the *directional radial basis functions* [19, RBFs], or the *Simple Harmonic Oscillator Based Reconstruction and Estimation* [20, SHORE]. More recently, the *Mean Apparent Propagator MRI* [12, MAP-MRI] and its improved version, the so-called *Laplacian-regularized MAP-MRI* [21, MAPL], have gained interest among the community due to the compelling results demonstrated in several clinical trials [22].

There is no doubt the EAP provides rich and valuable anatomical information about the diffusion process, though such amount of information may result overwhelming and difficult to integrate within clinical studies. This pitfall is usually circumvented by computing some sort of radial averaging of the EAP to obtain scalar measures directly related to the characteristics of diffusion. These measures act as biomarkers candidates aimed at describing diffusivity, anisotropy, intra-cellular vs. extra-cellular water movement, etcetera. Some prominent

examples in this sense are the probability of zero displacement (or return-to-origin probability, RTOP), the mean-squared displacement (MSD), the q-space inverse variance (QIV), or the return-to-plane and return-to-axis probabilities (RTPP, RTAP) [19, 23, 24].

Although the use of these measures is not generalized among the clinical community, there is a growing interest in the exploration of their potential clinical applicability. To date, the relevance of scalar descriptors of the brain micro-structure has been proved on both *ex vivo* [12, 25, 26] and *in vivo* studies of healthy and diseased subjects [22, 23, 27–30]. In particular, RTOP has also demonstrated to be a better indicator for cellularity and diffusion restrictions than the DTI-related mean diffusivity (MD) [22] and, together with MSD, a proper measure for the assessment of myelination [31]. These results were later confirmed by [29], where the authors reinforced the hypothesis on RTOP to have greater sensitivity to reflect cellularity and restricted diffusion.

The obvious drawback of this methodology is the need of acquiring very large data sets with many q-space samples in different shells (some of them with very large b-values, which implies an additional problem due to noise, eddy currents, non-linear effects, etcetera). Even when sophisticated non-linear techniques based on compressed sensing are used, the number of gradient images to be acquired vastly exceeds that needed for single-shell protocols like DT-MRI or HARDI. This is clearly a practical limitation: a large number of samples goes together with longer scanning times, subject movement, and patient discomfort that make them unfit for clinical practice and for many clinical studies. Besides, some methods require b-values that not every commercial MRI device is prepared to acquire.

The present paper delves into the question if scalar measures such as RTOP, RTPP, or RTAP are intrinsically tied up to the computation of the whole, model-free EAP. More precisely, we hypothesize that a constrained model for radial diffusion may reveal valuable information using simpler acquisition protocols, so that a set of *apparent* scalar measures probed at one single shell will exhibit a sensitivity to micro-structural changes comparable to *non-apparent* measures computed from the full EAP. The rationale behind this is that state of the art EAP techniques probe (instead of modeling) the actual radial behavior of the diffusion signal just to subsequently collapse it in a radial integral (average), so that the extra information provided by multi-shell acquisitions is indeed marginalized. In other words, we intend to substitute the whole average for all b-values with an *apparent* value at a single b-value.

To test our hypothesis, we have first reformulated RTOP, RTPP, and RTAP for a single-shell acquisition based on different diffusion models, yielding to closed form expressions and numerical implementations that are both robust and fast to compute. These *apparent* measures at one shell are compared with their state of the art counterparts based on the whole EAP in a set of experiments with real data sets. The figure of merit in such comparison is the ability to discern voxels with different anisotropy configurations, i.e., the sensitivity to micro-structural changes.

## Background

### The diffusion signal

The EAP, $P(\mathbf{R})$, is the three dimensional Probability Density Function (PDF) of the water molecules inside a voxel moving an effective distance $\mathbf{R}$ in an effective time $\tau$. It is related to the normalized magnitude image provided by the MRI scanner, $E(\mathbf{q})$, by the Fourier transform [32]:

$$P(\mathbf{R}) = \int_{\mathbb{R}^3} E(\mathbf{q})e^{-2\pi j\mathbf{q}\cdot\mathbf{R}}d\mathbf{q} = \mathfrak{F}\{|E(\mathbf{q})|\}(\mathbf{R}). \tag{1}$$

The inference of exact information on the **R**–space would require the sampling of the whole **q**–space to use the Fourier relationship between both spaces.

In order to obtain a closed-form analytical solution from a reduced number of acquired images, a model for the diffusion behavior must be adopted. The most common techniques rely on the assumption of a Gaussian diffusion profile and a steady state regime of the diffusion process that yields to the well-known Diffusion Tensor (DT) approach. Alternatively, a more general expression for $E(\mathbf{q})$ can be used [8]:

$$E(\mathbf{q}) = \exp(-4\pi^2\tau q_0^2 D(\mathbf{q})) = \exp(-b \cdot D(\mathbf{q})), \tag{2}$$

where the positive function $D(\mathbf{q}) = D(q_0, \theta, \phi) > 0$ is the Apparent Diffusion Coefficient (ADC), $b = 4\pi^2\tau\|\mathbf{q}\|^2$ is the so-called b-value and $q_0 = \|\mathbf{q}\|$, and $\theta, \phi$ are the angular coordinates in the spherical system. According to [33], in the mammalian brain, this mono-exponential model is predominant for values of b up to 2,000s/mm$^2$ and it can be extended to higher values (up to 3,000s/mm$^2$) if appropriate multi-compartment models of diffusion are used.

## Advanced diffusion scalar measures

Although the EAP provides the global information about the diffusion in every voxel of the brain, that information must be properly translated to be used in clinical trials or to study the features of particular tissues. Regardless of the method used to estimate the EAP, it must provide a set of metrics to inspect the changes of complex brain micro-structures, e.g., multiple compartments or restricted diffusion. Some of the most relevant measures usually derived from the EAP are:

1. **Return-to-origin probability (RTOP)**: also known as *probability of zero displacement*, it is related to the probability density of water molecules that minimally diffuse within the diffusion time $\tau$. It is known to provide relevant information about the white matter structure [23, 24, 34], and has demonstrated to be a better indicator for cellularity and diffusion restrictions than the DTI-related mean diffusivity (MD) [22]. It is defined as the value of $P(\mathbf{R})$ at the origin, related to the volume of the signal $E(\mathbf{q})$:

$$\mathrm{RTOP} = P(\mathbf{0}) = \int_{\mathbb{R}^3} E(\mathbf{q}) d\mathbf{q}. \tag{3}$$

2. **Return-to-plane probability (RTPP)**, defined as:

$$\mathrm{RTPP} = \int_{\mathbb{R}^2} P(\mathbf{R}_\perp) d\mathbf{R}_\perp = \int_{\mathbb{R}} E(q_\|) dq_\|, \tag{4}$$

where $q_\|$ denotes the direction of maximal diffusion. It is known to be a good indicator of restrictive barriers in the axial orientation, and it is related to the mean pore length [12].

3. **Return-to-axis probability (RTAP)**, defined as:

$$\mathrm{RTAP} = \int_{\mathbb{R}} P(R_\|) dR_\| = \int_{\mathbb{R}^2} E(\mathbf{q}_\perp) d\mathbf{q}_\perp, \tag{5}$$

where $\mathbf{q}_\perp$ is the set of directions perpendicular to $\mathbf{q}_\|$ (the one with maximal diffusion). It is also a directional scalar index and an indicator of restrictive barriers in the radial orientation. According to [12], RTPP and RTAP values can be seen as the *decomposition* of the RTOP values into two components, parallel and perpendicular to the maximum diffusion.

Remarkably, each one of these measures is computed in the **q**-space as an integral in either $\mathbb{R}$, $\mathbb{R}^2$, or $\mathbb{R}^3$, which in the spherical coordinates system translates to an integral over the radial coordinate $q \equiv \|\mathbf{q}\|$ that averages the measured signal $E(\mathbf{q})$ over all shells.

## Methods

### Diffusion measures from single shell acquisitions

The estimation of a given magnitude is always a trade-off between the available data and the complexity of the model. In this case, we consider a single shell acquisition compatible with HARDI: moderated-to-high b-value (ranging from 2,000s/mm$^2$ to 3,000s/mm$^2$) and moderated-to-large number of gradients. Since the amount of data is reduced, we are forced to assume a restricted diffusion model consistent with single-shell acquisitions: the ADC will be roughly independent from the radial direction within the range of b-values probed, so that $D(\mathbf{q}) = D(\theta, \phi)$. This way Eq (2) becomes:

$$E(\mathbf{q}) = E(q_0, \theta, \phi) = \exp(-4\pi^2\tau q_0^2 \, D(\theta, \phi)). \tag{6}$$

With this model, the radial integral in $q$ that defines all the previously introduced measures can be analytically computed without the need to actually sample $q$ itself. The corresponding formulations can be simplified accordingly:

1. **RTOP**: By using the simplification in Eq (6), we can write Eq (3) in spherical coordinates and integrate with respect to the radial component $q_0$:

$$
\begin{aligned}
\text{RTOP} &= \int_0^\infty \int_0^{2\pi} \int_0^\pi \exp(-4\pi^2\tau q_0^2 \cdot D(\theta, \phi))q_0^2\sin\theta \; d\phi \; d\theta \; dq_0 \\
&= \frac{1}{4}\frac{\sqrt{\pi}}{(4\pi^2\tau)^{3/2}} \int_S \frac{1}{D(\theta, \phi)^{3/2}} \, dS,
\end{aligned}
\tag{7}
$$

where $\int_S$ denotes the integral in the surface of a sphere $S$ of radius one. This way, the integration in the whole **q**-space in Eq (3) reduces to the integration on the surface of a single shell.

2. **RTPP**: The diffusion signal $D(\mathbf{q})$ in the maximum diffusion direction is given by $D(r_0)$, with $r_0 = q_\|$. Since that direction does not depend on $q_0$, we can integrate with respect to the radial component:

$$
\begin{aligned}
\text{RTPP} &= \int_{-\infty}^\infty \exp(-4\pi^2\tau q_0^2 D(r_0))dq_0 \\
&= \sqrt{\frac{\pi}{(4\pi^2\tau)}}\sqrt{\frac{1}{D(r_0)}}.
\end{aligned}
\tag{8}
$$

3. **RTAP**: Let $\theta'$ be the angle that parameterizes the equator normal to the maximum diffusion direction and $D(\theta')$ the diffusion signal at that equator. Once more, $D(\theta')$ does not depend on the radial component and the integral can be solved:

$$
\begin{aligned}
\text{RTAP} &= \int_0^\infty \int_0^{2\pi} \exp(-4\pi^2\tau q_0^2 D(\theta')) \; q_0 \; d\theta' \; dq_0 \\
&= \frac{1}{2 \cdot 4\pi^2\tau} \int_0^{2\pi} \frac{1}{D(\theta')} \, d\theta'.
\end{aligned}
\tag{9}
$$

The original integral reduces to the line integral of a function in a plane perpendicular to the maximum diffusion direction.

Although the mono-exponential assumption in Eq (6) may seem restrictive, it has been successfully adopted before for single-shell, HARDI models to accurately describe several predominant diffusion directions within the imaged voxel [7, 8, 35, 36]. Moreover, it allows to get rid of the dense sampling required by the original formulations of RTOP, RTPP, and RTAP, as long as the volumetric integrals over the whole q-space are replaced by surface integrals over one single shell.

On the other hand, the mono-exponential model will roughly hold only within a limited range around the measured b-value, but diffusion features will diverge for very different b-values. For this reason, the measures derived this way must be seen as *apparent* values at a given b-value, related to the original ones but dependent on the selected shell. In what follows, they will be referred to as **Apparent Measures Using Reduced Acquisitions (AMURA)**.

## Numerical implementation

We propose a robust numerical implementation of the integrals that define the *apparent* RTOP and RTAP, as well as the formula for the *apparent* RTPP, based on Spherical Harmonics (SH) expansions:

1. **RTOP**: the integral of a signal $H(\theta, \phi)$ over the surface of the unit sphere $S$ relates to the 0–th order coefficient (DC component) of its SH series expansion, $C_{0,0}\{H(\theta, \phi)\}$:

$$C_{0,0}\{H(\theta, \phi)\} = \frac{1}{\sqrt{4\pi}} \int_S H(\theta, \phi) dS, \tag{10}$$

   so that the RTOP becomes:

$$\text{RTOP} = \frac{1}{(4\pi)^2 \tau^{3/2}} C_{0,0}\{(D(\theta, \phi))^{-3/2}\}. \tag{11}$$

2. **RTPP**: The value of RTPP previously defined in Eq (8) depends on $D(r_0)$, the ADC evaluated at the direction of maximum diffusion. In order to avoid the variability that a maximum operator may introduce, we calculate the index over a regularized version of $D(\theta, \phi)$. Let us call $D_{\text{SH}}(\theta, \phi)$ a version of the original diffusion signal regularized using SH. Then, we can write the RTPP as

$$\text{RTPP} = \frac{1}{\sqrt{4\pi\tau}} \frac{1}{\sqrt{D_{\text{SH}}(\mathbf{r}_0)}}, \tag{12}$$

   where $\mathbf{r}_0$ denotes the maximum diffusion direction.

3. **RTAP**: The value of $\int_0^{2\pi} D(\theta')^{-1} d\theta'$ is the line integral of $D(\theta')^{-1}$ along an equator perpendicular to the direction of maximum diffusion $\mathbf{r}_0$, i.e., the Funk-Radon Transform (FRT) of $D(\theta')^{-1}$ evaluated at $\mathbf{r}_0$, $\mathcal{G}\{D\}(\mathbf{r}_0)$ [37]:

$$\text{RTAP} = \frac{1}{2 \cdot 4\pi^2 \tau} \mathcal{G}\left\{\frac{1}{D(\theta')}\right\}(\mathbf{r}_0) = 2\Psi(\mathbf{r}_0), \tag{13}$$

   where $\Psi(\mathbf{r})$ is the pQ-Balls whose definition and SH-based numerical implementation are addressed in [38, 39].

**Table 1. Survey of the q-space measures gathered by AMURA, along with their specific numerical implementations.**

| Measure | Definition | Numerical Implementation |
|---|---|---|
| RTOP | $= \dfrac{1}{4} \dfrac{\sqrt{\pi}}{(4\pi^2\tau)^{3/2}} \int_S \dfrac{1}{D(\theta,\phi)^{3/2}} \, dS$ | $= \dfrac{1}{(4\pi)^2 \tau^{3/2}} C_{0,0} \{ (D(\theta,\phi))^{-3/2} \}$ |
| RTPP | $= \dfrac{1}{\sqrt{4\pi\tau}} \dfrac{1}{\sqrt{D(\mathbf{r}_0)}}$ | $= \dfrac{1}{\sqrt{4\pi\tau}} \dfrac{1}{\sqrt{D_{SH}(\mathbf{r}_0)}}$ |
| RTAP | $= \dfrac{1}{2 \cdot 4\pi^2\tau} \int_0^{2\pi} \dfrac{1}{D(\theta')} \, d\theta'$ | $= 2\Psi(\mathbf{r}_0)$, see [39] |

An overview of AMURA, together with the specific numerical implementation of each *apparent* measure, is presented in Table 1.

## Experiments and results

### Setting-up of the experiments

As explained above, AMURA measures rely on the expansion of spherical functions at a given shell in the basis of SH, for which the implementation described in [40] is used: even SH orders up to 6 are fitted with a Laplace-Beltrami penalty $\lambda = 0.006$. RTAP is computed from pQ-Balls with this same design for SH expansions [39]. For the sake of repeatability, *in vivo* data have been chosen exclusively from publicly available databases:

1. From the Human Connectome Project (HCP), five volumes were chosen: MGH 1007, MGH 1010, MGH 1016, MGH 1018 and MGH 1019, acquired in a Siemens 3T Connectome scanner with 4 different shells at b = {1,000,3,000,5,000,10,000} s/mm$^2$, with {64, 64, 128, 256} gradient directions each, in-plane resolution 1.5 mm$^2$, and slice thickness 1.5 mm. Acquisition parameters are TE = 57 ms and TR = 8800 ms. These data were obtained from the Human Connectome Project (HCP) database (https://ida.loni.usc.edu/login.jsp). The HCP project (Principal Investigators: Bruce Rosen, M.D., Ph.D., Martinos Center at Massachusetts Gen eral Hospital; Arthur W. Toga, Ph.D., University of Southern California, Van J. Weeden, MD, Martinos Center at Massachusetts General Hospital) is supported by the National Institute of Dental and Craniofacial Research (NIDCR), the National Institute of Mental Health (NIMH) and the National Institute of Neurological Disorders and Stroke (NINDS). HCP is the result of efforts of co-investigators from the University of Southern California, Martinos Center for Biomedical Imaging at Massachusetts General Hospital (MGH), Washington University, and the University of Minnesota.
   This acquisition included 40 different baselines that were averaged to improve their SNR. The SNR of each of the individual baselines is high enough to make a Gaussian approximation feasible with a small error. Under this approximation we can assure that the average operator provides an unbiased output image [41].

2. From the Public Parkinson's Disease database (PPD), 53 subjects from a cross-sectional Parkinson's Disease (PD) study comprising 27 patients together with 26 age, sex, and education-matched control subjects. Data were acquired on a 3T head-only MR scanner (Magnetom Allegra, Siemens Medical Solutions, Erlangen, Germany) operated with an 8-channel head coil. Diffusion-weighted (DW) images were acquired with a twice-refocused, spin-echo sequence with EPI readout at two distinct b-values b = {1,000, 2,500} s/mm$^2$, and along 120 evenly spaced encoding gradients. For the purposes of motion correction, 22 unweighted (b = 0) volumes, interleaved with the DW images, were acquired. Acquisition parameters are TR = 6800 ms, TE = 91 ms, and FOV = 211 mm$^2$, no parallel

imaging and 6/8 partial Fourier were used. More information can be found in [42]. These data were acquired at the Cyclotron Research Centre, University of Liège. Available: https://www.nitrc.org/frs/?group_id=835.

## Consistency of *apparent*, single-shell measures

Since AMURA are intended to reveal similar micro-structural changes as multi-shell EAP estimators, each one of the *apparent* RTOP, RTPP, and RTAP are expected to correlate well with their multi-shell counterparts, meaning the anatomical information they assess is closely related. To check this point, AMURA is compared against three state of the art EAP estimation techniques not requiring dense samplings of the q-space: RBFs with constrained $\ell_2$ regularization as described in [19], MAP-MRI with anisotropic basis and radial order 6 [12], and MAPL with anisotropic basis, radial order 8, and regularization weighting $\lambda = 0.2$ [21].

In order to attain an affordable complexity for this experiment, the study is restricted to three different axial slices for each selected volume as depicted in Table 2.

For each volume and slice, the three measures are calculated with RBFs, MAP-MRI, and MAPL using either 3 shells (b = {1,000, 3,000, 5,000} s/mm$^2$), or 2 shells (b = {1,000, 3,000} s/mm$^2$). AMURA are calculated using one single shell at either b = 3,000s/mm$^2$ or b = 5,000s/mm$^2$. This sum up 8 different calculations of each of RTOP, RTPP and RTAP for each volume and slice as illustrated in Fig 1, where those voxels with FA bellow 0.2 have been masked.

A simple visual inspection suggests that indeed all the 8 different computations of RTOP, RTPP, and RTAP provide congruent information about the anatomies imaged. This qualitative evidence is confirmed in Table 3, where the correlation coefficients $\rho$ between each pair of measures are computed. In precise terms, let $\{r_i\}_{i=1}^{N}$ be the values of the measure defined at each row of Table 3, and $\{c_i\}_{i=1}^{N}$ the values of the measure defined at each column; the set $i = 1 \ldots N$ gathers all those voxels with FA above 0.2. Then:

$$\rho_{rc} = \frac{\sum_{i=1}^{N}(r_i - \bar{r})(c_i - \bar{c})}{(N-1)\sigma_r \, \sigma_c}, \ \text{ for}: \ \bar{x} = \frac{1}{N}\sum_{i=1}^{N}x_i \ \text{ and } \ \sigma_x = \frac{1}{N-1}\sum_{i=1}^{N}(x_i - \bar{x})^2. \tag{14}$$

Results for RTOP show a strong correlation, in some cases over 90%, between the measure estimated with AMURA and the calculation given by the other techniques, particularly those based on MAP. It is worth noticing that AMURA-RTOP correlates better with MAP-RTOP than RBF-RTOP does, even when RBF is computed from 3 shells (left column) and AMURA is using as few as 64 gradients (b = 3,000s/mm$^2$) or 128 gradients (b = 5,000s/mm$^2$) in one single shell.

For RTPP, though the absolute correlations between each pair of computations are clearly weaker than for RTOP, AMURA still exhibits a higher consistency towards MAP-based measures than RBF does. At the sight of Fig 1, the noisier nature of RTPP could probably explain

**Table 2. Selected slices from each diffusion volume from the HCP.**

| Volume | Slice numbers | Volume | Slice numbers |
|---|---|---|---|
| MGH 1007 | 42, 52, 65 | MGH 1018 | 31, 41, 51 |
| MGH 1010 | 46, 54, 60 | MGH 1019 | 40, 50, 64 |
| MGH 1016 | 42, 55, 68 | | |

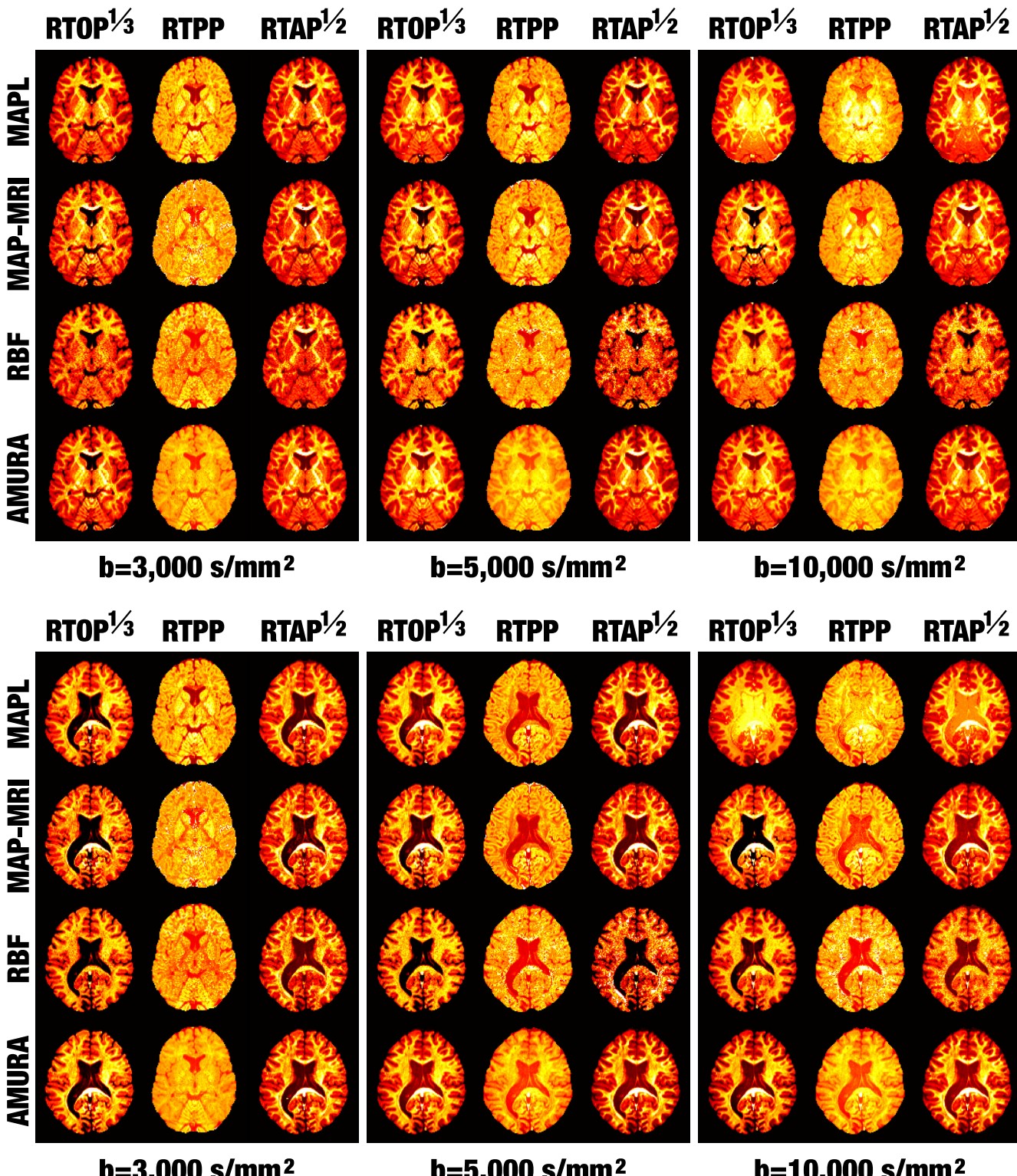

**Fig 1. Visual assessment of the consistency of AMURA.** (Top) Slice 42 of the MGH1016 volume from HCP; (Bottom) Slice 51 of the MGH1018. AMURA is calculated with one single shell at the specified b-value. MAPL, MAP-MRI and RBF are calculated using either 2 or 3 shells at the maximum b-value specified. For the sake of visual comparison, RTOP and RTAP have been gamma-corrected as specified.

**Table 3. Correlation coefficients between the different methods to estimate RTOP, RTPP and RTAP.** The higher the better. AMURA are computed from one single shell at either b = 3,000s/mm² (3k) or b = 5,000s/mm² (5k). Multi-shell methods are always compared between them with the same number of shells (2 or 3).

| | | 3 shells | | | 2 shells | | |
|---|---|---|---|---|---|---|---|
| | | **RBF** | **MAPL** | **MAP-MRI** | **RBF** | **MAPL** | **MAP-MRI** |
| RTOP | **AMURA** 3k | 0.7636 | 0.8616 | 0.9202 | 0.8051 | **0.9047** | **0.9027** |
| | **AMURA** 5k | 0.7629 | **0.9538** | **0.9151** | 0.7264 | 0.8950 | 0.8278 |
| | **RBF** | – | 0.7320 | 0.6408 | – | 0.7746 | 0.7136 |
| | **MAPL** | – | – | 0.8356 | – | – | 0.7334 |
| RTPP | **AMURA** 3k | 0.2565 | **0.7035** | 0.6811 | 0.6464 | **0.7497** | 0.6423 |
| | **AMURA** 5k | 0.2295 | 0.6077 | 0.4530 | 0.3155 | 0.3884 | 0.2415 |
| | **RBF** | – | 0.1041 | 0.1139 | – | 0.7089 | 0.6096 |
| | **MAPL** | – | – | **0.9416** | – | – | **0.8678** |
| RTAP | **AMURA** 3k | 0.4918 | 0.8800 | **0.9305** | 0.7846 | 0.8955 | **0.9341** |
| | **AMURA** 5k | 0.5145 | 0.9382 | **0.9406** | 0.8009 | 0.8993 | **0.9049** |
| | **RBF** | – | 0.4740 | 0.4706 | – | 0.7739 | 0.8170 |
| | **MAPL** | – | – | 0.8885 | – | – | 0.8451 |

the net decrease in the correlations. Interestingly, the computation of RTPP with 2 shells seems more consistent between multi-shell techniques than it is with 3 shells. For example, the correlation between RBF-RTPP and MAPL-RTPP falls as low as 10%.

Since RTAP provides *cleaner* maps than RTPP (see Fig 1), the discussion becomes similar to the case of RTOP: the overall correlations between the different computations are much higher in this case, with AMURA correlating up to 90% with MAPL and MAP-MRI. Once again, RBF-RTAP seems less consistent with MAP-like-RTAP than AMURA-RTAP.

Summarizing, AMURA provide information that closely resembles that computed with multi-shell methods. Moreover, AMURA are more consistent with MAP-like measures than other multi-shell methods like RBF. This might suggest that the deviations introduced by the election of different basis functions and different numerical schemes in each multi-shell method could indeed surpass the error AMURA introduce as a consequence of modeling (instead of sampling) the radial behavior of $E(\mathbf{q})$.

### Sensitivity of *apparent* single-shell measures to tissue properties

Though AMURA provide anatomical maps that closely resemble those yielded by multi-shell methods (see Fig 1), it is not necessarily implied that they have the same capabilities to distinguish analogous tissue properties. Such capabilities are first put to the test by means of a classification problem where two classes are defined depending on the values of either RTOP, RTPP, or RTAP computed from MAPL with 4 shells, see Fig 2. This way, MAPL becomes a bronze standard given its high consistency with both MAP-MRI and AMURA (it shows also the strongest correlations with RBF, see Table 3), and assuming it probes actual micro-structural information. The problem design is as follows:

1. Once the background of the image is removed, the histogram of either RTOP, RTPP, or RTAP is computed from the bronze standard (MAPL). A threshold is selected in the valley right after the main lobe for each MAPL-measure (for RTOP: $2 \cdot 10^6 \text{mm}^{-3}$; for RTPP: $90 \text{mm}^{-1}$; for RTAP: $1.5 \cdot 10^4 \text{mm}^{-2}$). Classes 1 and 2 are defined as either below or above this threshold, see Fig 2(A).

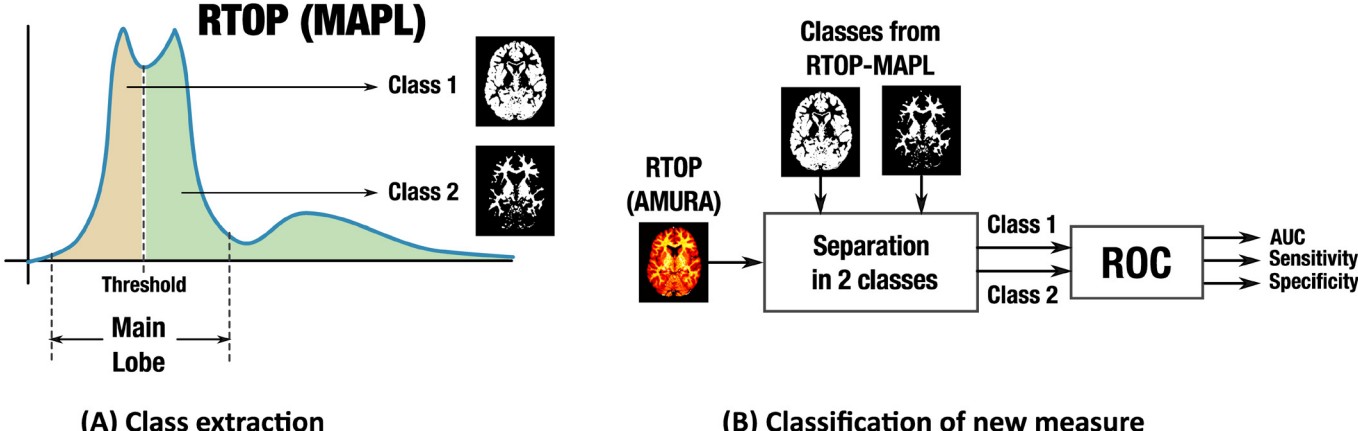

**Fig 2. Conceptual description of the problem designed to test the sensitivity of AMURA to micro-structural changes.** (A) The pixels in the image are split into 2 classes by thresholding the corresponding MAPL measure (RTOP in the example). (B) Each one of the methods to be tested: MAPL, MAP-MRI, RBF, or AMURA (AMURA in the example) is used to compute this same measure, and a ROC curve is calculated with the classes defined in (A) as the target.

2. From each of the other methods (MAP-MRI, RBF, AMURA, and MAPL itself with less that 4 shells), RTOP, RTPP, and RTAP are computed and used as discriminant features of each voxel.

3. In case a given method were actually providing the exact same micro-structural information as the bronze standard, such features should suffice to mimic the exact same classification designed in Fig 2(A). Otherwise, both false positives (class 1 voxels tagged as class 2) and false negatives (class 2 voxels tagged as class 1) will appear that reflect discrepancies in the information measured.

4. Such discrepancies are quantified by means of a Receiver Operating Characteristics (ROC) curve: for a given measure, corresponding values are computed using each method; these values are further classified using a *moving* threshold ranging from the minimum computed value to its maximum. This way, each value of the *moving* threshold defines a classification that is compared to the bronze standard in step 1 in search for false positives and false negatives. The ROC curve is the graphic relating these two rates as the threshold moves. Finally, three standard Figures of Merit (FoM) related to the ROC are reported: the area under the curve (AUC), the sensitivity at the optimum threshold, and the specificity at the optimum threshold, see Fig 2(B).

The results are gathered, respectively, in Table 4 (RTOP), Table 5 (RTPP), and Table 6 (RTAP). In all cases, the closer to 1 is the better. While AMURA are computed from one shell, the other methods use either 2 shells (at maximum b = 3,000s/mm$^2$), 3 shells (at maximum b = 5,000s/mm$^2$) or 4 shells (at maximum b = 10,000s/mm$^2$).

As can be seen, AMURA scores high FoMs in all cases, even above those obtained with MAP-MRI (which is a non-improved version of MAPL itself). For example, the *apparent* value of AMURA-RTOP at any shell scores higher than any of the computations from MAP-MRI regardless on the number of shells it uses (Table 4). Indeed, this same comment holds true for the other two measures, with the exception of the specificity of RTPP with MAP-MRI at 4 shells (Table 5) and the specificity of RTAP with MAP-MRI at 4 shells (Table 6). In the same way as in Table 3, the measures computed with RBF tend to deviate from those based on MAP

**Table 4. ROC FoMs for RTOP (the closer to 1, the better).** MAPL with 4 shells at a maximum b = 10,000s/mm$^2$ is the bronze standard.

| AUC | | MAPL | AMURA | RBF | MAP-MRI |
|---|---|---|---|---|---|
| | b = 3000 | 0.8796 | 0.8285 | 0.6887 | 0.6839 |
| | b = 5000 | **0.9343** | **0.9205** | 0.7251 | 0.7035 |
| | b = 10000 | **1.0000** | **0.9771** | 0.7762 | 0.7219 |
| Sensitivity | | MAPL | AMURA | RBF | MAP-MRI |
| | b = 3000 | 0.8114 | 0.7527 | 0.6108 | 0.6123 |
| | b = 5000 | **0.8802** | **0.8520** | 0.6480 | 0.6378 |
| | b = 10000 | **1.0000** | **0.9213** | 0.7318 | 0.6402 |
| Specificity | | MAPL | AMURA | RBF | MAP-MRI |
| | b = 3000 | **0.9114** | 0.8367 | 0.7915 | 0.8109 |
| | b = 5000 | **0.9454** | **0.9334** | 0.8480 | 0.8285 |
| | b = 10000 | **1.0000** | **0.9623** | **0.9359** | 0.8788 |

**Table 5. ROC FoMs for RTPP (the closer to 1, the better).** MAPL with 4 shells at a maximum b = 10,000s/mm$^2$ is the bronze standard.

| AUC | | MAPL | AMURA | RBF | MAP-MRI |
|---|---|---|---|---|---|
| | b = 3000 | 0.7884 | 0.6900 | 0.5736 | 0.5550 |
| | b = 5000 | **0.8647** | 0.7657 | 0.4632 | 0.6038 |
| | b = 10000 | **1.0000** | **0.8261** | 0.4735 | 0.6488 |
| Sensitivity | | MAPL | AMURA | RBF | MAP-MRI |
| | b = 3000 | 0.6761 | 0.5803 | 0.5008 | 0.5000 |
| | b = 5000 | **0.7516** | 0.6332 | 0.4807 | 0.5295 |
| | b = 10000 | **1.0000** | **0.7077** | 0.4828 | 0.5677 |
| Specificity | | MAPL | AMURA | RBF | MAP-MRI |
| | b = 3000 | **0.7828** | 0.7162 | 0.7260 | 0.6608 |
| | b = 5000 | **0.8440** | 0.7469 | 0.5171 | 0.7442 |
| | b = 10000 | **1.0000** | **0.7713** | 0.5440 | **0.8284** |

even if the number of shells increases. Finally, it is worth noticing that the *apparent* values obtained with AMURA at either b = 3,000s/mm$^2$ or b = 5,000s/mm$^2$ score pretty close to MAPL when the outermost shell at b = 10,000s/mm$^2$ is suppressed from the bronze standard.

Summarizing, not only AMURA strongly correlate with measures derived from multi-shell techniques, but they seem to distinguish tissue properties as well as the other methods do.

**Table 6. ROC FoMs for RTAP (the closer to 1, the better).** MAPL with 4 shells at a maximum b = 10,000s/mm$^2$ is the bronze standard.

| AUC | | MAPL | AMURA | RBF | MAP-MRI |
|---|---|---|---|---|---|
| | b = 3000 | **0.9218** | **0.8959** | 0.7338 | 0.7592 |
| | b = 5000 | **0.9537** | **0.9446** | 0.6543 | 0.7717 |
| | b = 10000 | **1.0000** | **0.9755** | 0.7456 | 0.7993 |
| Sensitivity | | MAPL | AMURA | RBF | MAP-MRI |
| | b = 3000 | 0.8516 | 0.8204 | 0.6309 | 0.6844 |
| | b = 5000 | **0.8864** | **0.8808** | 0.5911 | 0.6997 |
| | b = 10000 | **1.0000** | **0.9223** | 0.6900 | 0.7430 |
| Specificity | | MAPL | AMURA | RBF | MAP-MRI |
| | b = 3000 | **0.9232** | 0.8848 | 0.8266 | 0.8473 |
| | b = 5000 | **0.9480** | **0.9152** | 0.6665 | 0.8612 |
| | b = 10000 | **1.0000** | **0.9482** | 0.7728 | **0.9205** |

Interestingly, the micro-structural properties described by multi-shell techniques do not seem to *converge* even if the q-space sampling is improved.

## Potential of *apparent* single-shell measures in clinical setups

The previous experiment relies on an artificial classification of voxels depending on MAPL as a bronze standard. To further test the capabilities of AMURA to probe tissue properties, we have devised an additional experiment involving the clinical data in the PPD database. Though PD is known to affect the substantia nigra or the gray matter more than the *standard* white matter tracts commonly studied in group-wise analyses based on DMRI, significant differences have also been reported in several white matter regions such as the corpus callosum, the corticospinal tract, or the fornix [43]. Accordingly, we have focused on commonly-studied white matter tracts that are segmented for each volume in the PDD database based on the ENIGMA-DTI template [44] (ENIGMA project web page: http://enigma.ini.usc.edu/; template data and processing protocols for DTI: https://www.nitrc.org/projects/enigma_dti) and the JHU WM atlas [45] as follows:

1. The FA is calculated as a reference value using MRTRIX (http://www.mrtrix.org) for $b = 1,000s/mm^2$. Its value is registered against the ENIGMA-DTI FA template using deformable image registration based on the local cross-correlation between the images [46].

2. The JHU WM atlas classifies 48 disjointed white matter regions in the image space of the ENIGMA-DTI template. Their segmentations are back-projected onto the image space of each subject in the PDD database using the output deformation field of the registration. Working on the original image space avoids interpolation artifacts as well as side effects induced by the higher resolution of the ENIGMA-DTI template as compared to the PDD subjects.

3. The ENIGMA-DTI template comprises segmentations of both the whole white matter tracts and their FA skeletons. Back-projection is repeated for both segmentations, hence both a full segmentation of each tract and its pseudo-skeleton (central core) is available in the original image space (see Fig 3).

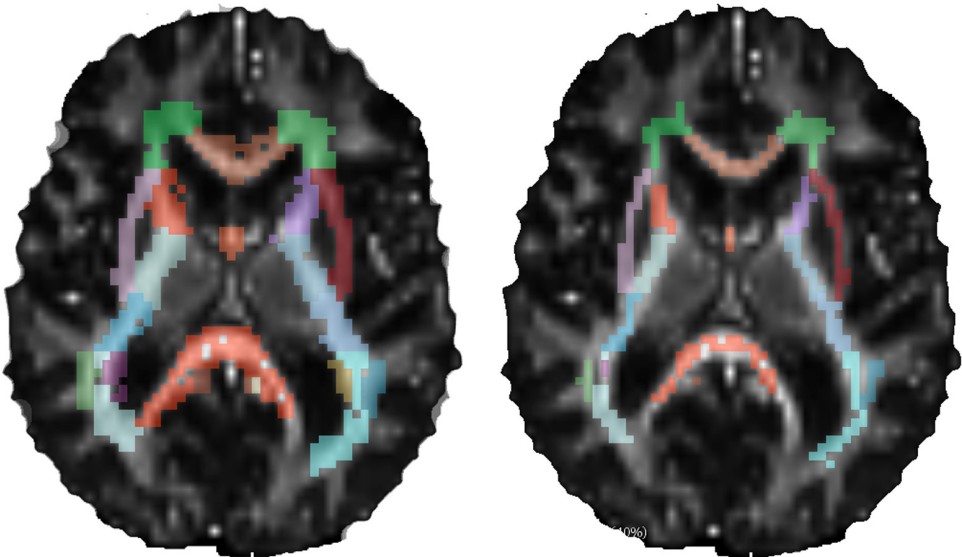

**Fig 3. Registration-based segmentation of WM tracts of a control subject in the PDD database.** (Left) Whole tracts. (Right) Pseudo-skeletons.

4. Outliers are removed from the segmentations by eliminating those voxels with abnormal values (i.e. values outside the range $[0, 1]$) of the FA and "Westin's scalars", $C_p$, $C_l$, $C_s$ [5].

Each segmented tract is characterized by one single scalar measure: for AMURA, the *apparent* RTOP, RTPP, and RTAP at b = 2,500s/mm$^2$ are averaged over each pseudo-skeleton. As in the previous section, their MAPL counterparts (using the 2 available shells) are targeted to as the state of the art. Additionally, a tensor model-driven version of the indices (at b = 2,500s/mm$^2$) is tested as a sort of end of scale (see Appendix for the implementation details). Finally, the raw FA is also included in the analysis since it is the standard index to test in group studies [43].

Among the 48 tracts in the JHU WM, we have found statistically relevant differences mainly at the corpus callosum, which is in agreement with the related literature [43]. Table 7 shows the results for two-sample, pooled variance *t*-tests over Gaussian-corrected data between controls and patients for each of the measures considered and at each of the three sections of the corpus callosum segmented in the JHU WM (genu –GCC–, body –BCC–, and splenium – SCC–).

RTPP-related measures result in discriminant markers for this particular problem at the genu and the splenium of the corpus callosum. Remarkably, the raw FA is only able to find differences at the splenium, meanwhile RTAP and RTOP are unable to plot significant differences in a consistent way. To further understand why RTPP consistently finds significant differences, and how this is related to the information it measures, its actual distribution (PDF) inside the pseudo-skeleton of each segment (GCC, BCC, SCC) is estimated by using Parzen windowing in Fig 4.

AMURA-RTPP is able to consistently distinguish between two different populations within each region of the corpus callosum. Meanwhile these two groups are also discriminated at the genu by the other approaches, this is not the case at the body and, above all, at the splenium, where even the MAPL-RTPP fails to find the valley between the two populations. Specifically, statistically significant differences between controls and patients appear wherever there is a change in the relative distribution of voxels between the two populations, i.e., at both the genu and the splenium. This provided, and anytime the separation between the two populations can be easily identified at AMURA − RTPP = 27mm$^{-1}$, the segmentation of the corpus callosum depending on its *apparent* RTPP is straightforward by thresholding. Such processing has been applied to each subject in the database (both controls and patients), and the resulting

**Table 7. Two-sample *t*-tests for each measure and at each section of the corpus callosum (the lower the better).** The *p*-values represent the probability that the measure has identical means for both controls and patients. Differences with statistical significance above 99% (resp. 95%) are highlighted in green (resp. amber).

| | Tensor | Tensor | | | MAPL | | |
|---|---|---|---|---|---|---|---|
| | FA | RTOP | RTPP | RTAP | RTOP | RTPP | RTAP |
| GCC | 0.087 | 0.357 | 0.028 | 0.174 | 0.557 | 0.021 | 0.322 |
| BCC | 0.055 | 0.165 | 0.749 | 0.130 | 0.334 | 0.420 | 0.172 |
| SCC | 0.014 | 0.135 | 0.036 | 0.030 | 0.164 | 0.015 | 0.069 |

| | AMURA | | |
|---|---|---|---|
| | RTOP | RTPP | RTAP |
| GCC | 0.334 | 0.011 | 0.214 |
| BCC | 0.193 | 0.470 | 0.137 |
| SCC | 0.272 | 0.003 | 0.144 |

**Fig 4. PDFs of RTPP computed from either the tensor model, MAPL, or AMURA (plus the FA) and within each of GCC, BCC, or SCC.** (Red) Patients; (Green) Controls. (Dashed line) Bootstrap PDF from 100 iterations with 15 subjects each; (Solid line) Global PDF for all controls/patients. The *p*-values are referred to the *t*-tests reported in Table 7.

segmentations have been projected onto the image space of the ENIGMA template to compute the average segmentation shown in Fig 5.

The two populations identified by AMURA-RTPP correspond to a clean segmentation of the corpus callosum distinguishing between its lowermost (closer to the cerebrospinal fluid) and its uppermost (closer to the cingulum) sections, so that we can reasonably argue that

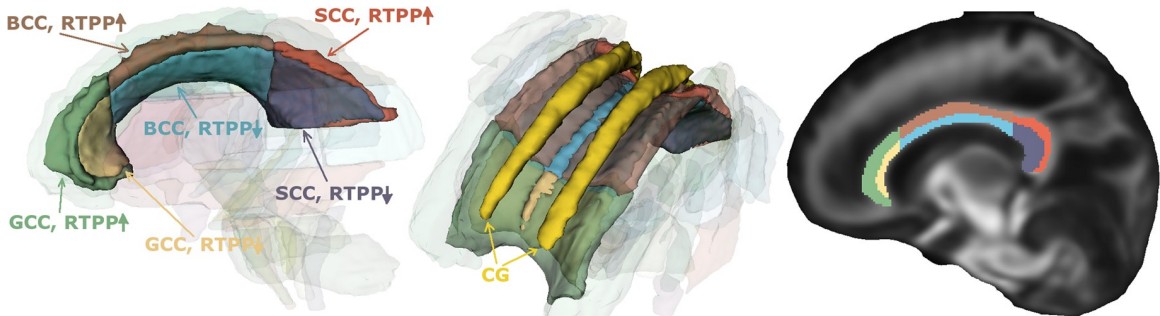

**Fig 5. Average segmentation of the corpus callosum in the space of the ENIGMA template by AMURA-RTPP thresholding at 27 mm$^{-1}$.** The cingulum (CG) is also rendered in the 3D view for reference purposes. A sagittal slice of the average FA of the PDD is also shown for reference.

AMURA-RTPP is actually able to discern micro-structural properties that remain hidden with DT-related measures (see Fig 4).

## Variability of *apparent* measures depending on the acquisition parameters

Since AMURA provide *apparent* measures at a given shell, the question of how much these measures depend on the actual shell measured naturally arises. As long as AMURA have been designed for *reduced* acquisition protocols, it also makes sense to check their sensitivity to the number of diffusion samples taken at a given shell. To put this to the test, a set of experiments have been designed using volume MGH 1016: the variability with the b-value is probed by subsequently computing AMURA with each of the available shells at either b = 3,000s/mm$^2$, b = 5,000s/mm$^2$, or b = 10,000s/mm$^2$. For the variability with the number of diffusion gradients, we start with the 128 samples at b = 5,000s/mm$^2$ and uniformly subsample this set to obtain either 32, 48, 64, 80, 96, 112, or 128 diffusion directions subsets (a "uniform" subsampling of *n* gradients among the original 128 is here defined as those *n* directions that minimize the overall electrostatic repulsion energy amongst all $\binom{128}{n}$ combinations. The optimization is carried out using heuristic rules). To plot such a huge amount of information, only those voxels of MGH 1016 with FA above 0.2 are included, and they are further clustered depending on their FA using fuzzy c-means. This results in 5 classes with centroids $C_L$ = {0.24, 0.36, 0.51, 0.66, 0.86}, for which the median of each AMURA measure is used as a representative, see Fig 6. AMURA seem extremely robust to the number of acquired gradients even in the case of very heavy subsamplings. This is as expected, since Fig 6 shows mean values but not variances. On the contrary, all three measures show a clear dependency with the b-value since the assumption that $D(\theta, \phi)$ is roughly constant holds only within a limited range of b-values. In any case, the monotonical behavior of each cluster is preserved for both RTOP and RTAP, i.e.

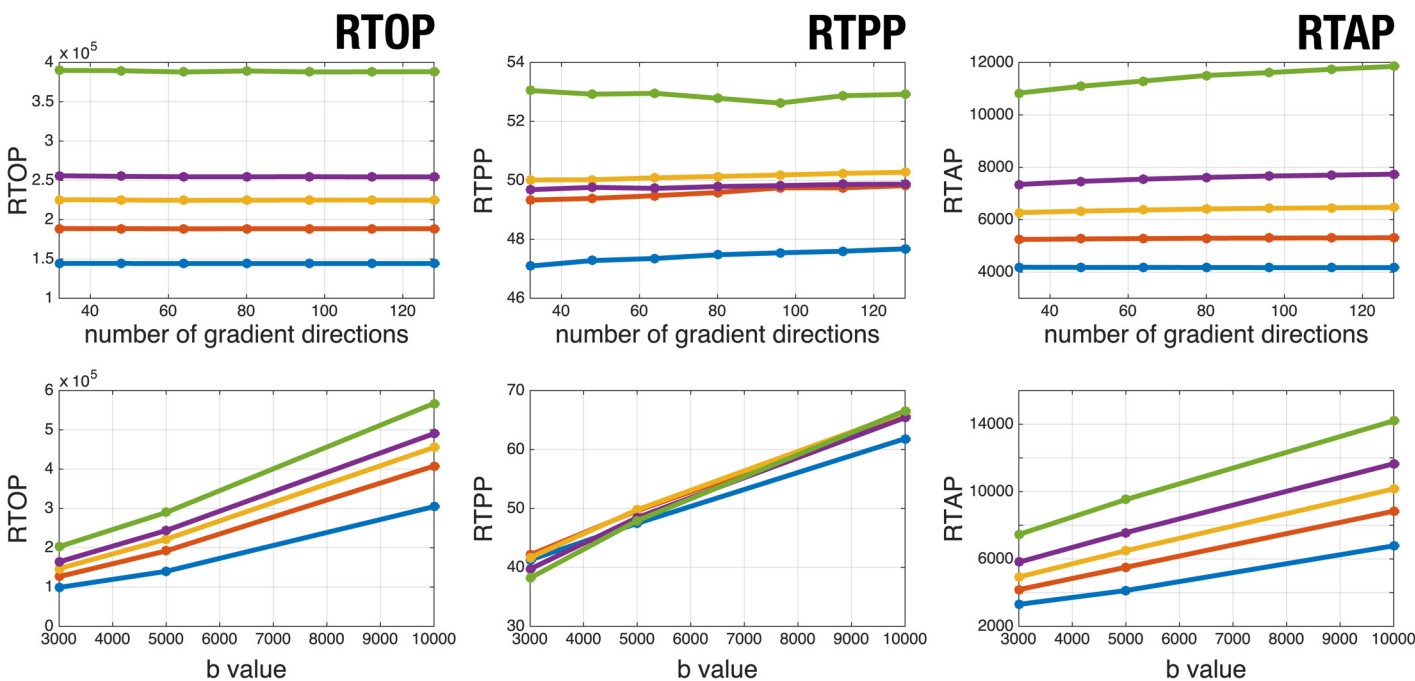

**Fig 6.** *Apparent* **values of AMURA as a function of the number of acquired gradients (top) or the b-value (bottom) for subject MGH 1016.** Each line correspond to a cluster of FA values computed from 5-fold fuzzy c-means. AMURA as a function of the number of gradients (top) are depicted at b = 5,000s/mm$^2$.

# RTPP

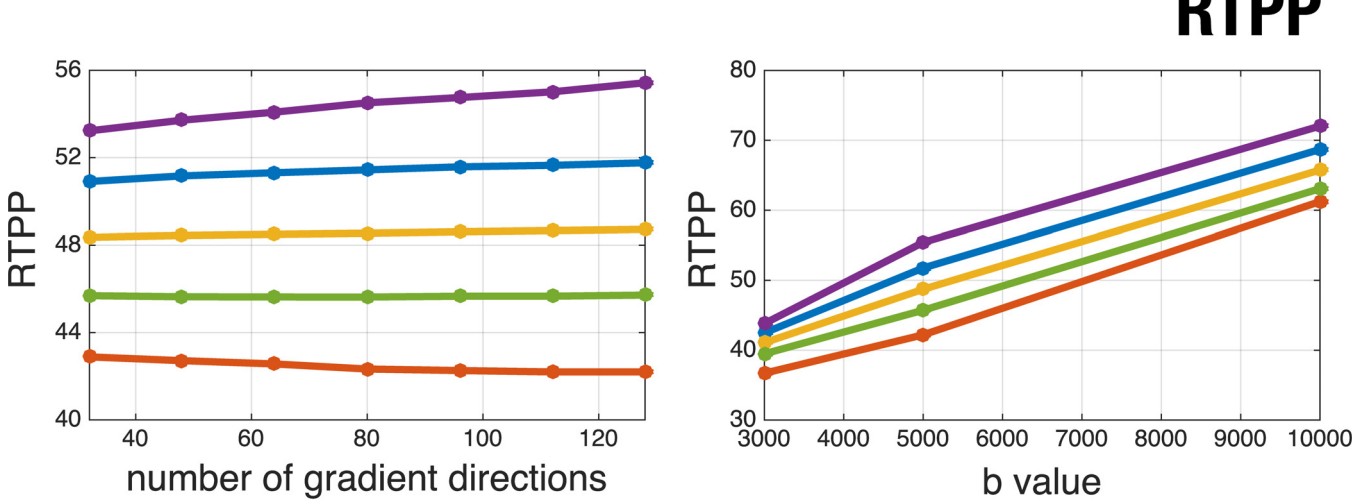

**Fig 7. AMURA-RTPP as a function of the number of acquired gradients (left) or the b-value (right) for subject MGH 1016.** Each line correspond to a cluster of RTPP values computed from 5-fold fuzzy c-means. AMURA-RTPP as a function of the number of gradients (left) is depicted at b = 5,000s/mm$^2$.

an increasing value of the FA comes along with an increasing value of RTOP and RTAP for all shells. Since both RTOP and RTAP resemble anisotropy maps, see Fig 1, this is as expected. This is not necessarily the case for RTPP, whose graphics for each cluster cross each other as the b-value varies. If the experiment is repeated for RTPP using a clustering of its own (i.e., by running fuzzy c-means over RTPP itself at b = 5,000s/mm$^2$, yielding five centroids $C_L$ = {20.72, 22.80, 24.35, 25.92, 27.83}), a perfect monotonical behavior is of course obtained as shown in Fig 7.

A similar test may be run over the multi-shell techniques. In this case we are interested in checking the variability of the measures depending on the number of shells used (either 2, 3, or 4). The same five volumes and 3 slices in Table 2 are used, and the fuzzy c-means clustering above described is repeated yielding centroids $C_L$ = {0.24, 0.33, 0.45, 0.58, 0.76}. Fig 8 demonstrates that indeed multi-shell measures do depend on the sampling scheme (number of shells).

Specifically, including the fourth shell at b = 3,000s/mm$^2$ heavily alters the measured RTOP, RTPP, and RTAP in all cases. Note that, while the monotonical behavior of RTOP and RTAP holds for MAP-like estimators, this is not always the case for RBF (which, in the light of this experiment, seems particularly unstable). As it was pointed out in the previous paragraph, RTPP is not necessarily expected to monotonically increase with the FA in any case.

## Computational issues and execution times

AMURA relies on SH expansions computed as linear, regularized LS problems. On the contrary, multi-shell methods depend on heavily non-linear, sparsity-driven, possibly constrained optimization problems. The linear nature of LS usually yields to well-behaved, stable solutions, meanwhile non-linear optimization usually arises numerical issues.

Besides, the computational load of LS is noticeably more modest (it reduces to invert one single matrix for the whole volume or even the whole cohort), to the point that AMURA can be several orders of magnitude faster than multi-shell techniques. This is illustrated here through volume MGH 1016 from the HCP. The measures of interest are computed on a quad-core Intel(R) Core(TM) i7-6700K 4.00GHz processor under Debian GNU/Linux 8.6 SO. The

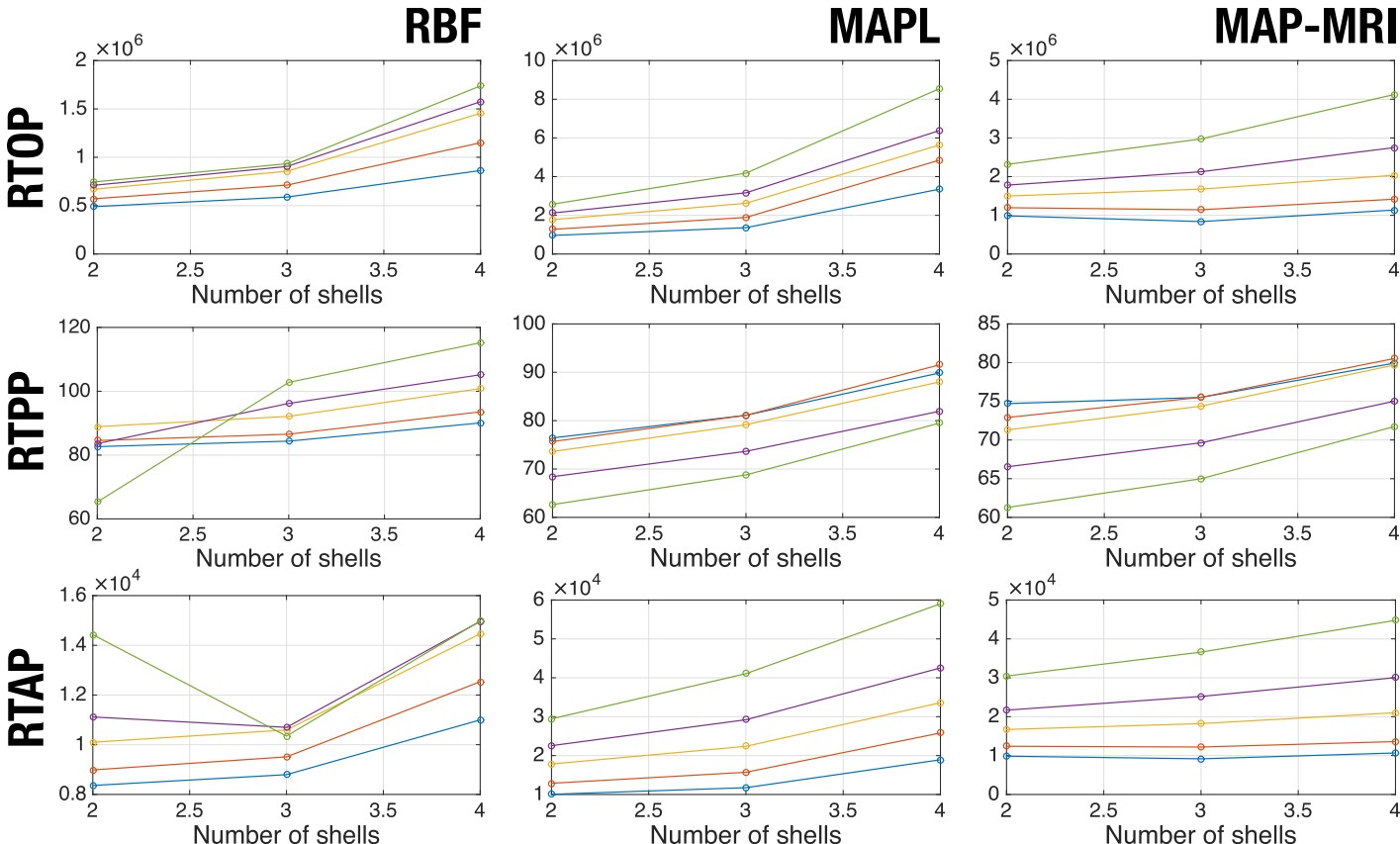

**Fig 8. Measured values with multi-shell techniques as a function of the number of shells acquired.** Each line correspond to a cluster from 5-fold fuzzy c-means.

available code for RBFs (https://github.com/LipengNing/RBF-Propagator) was run under MATLAB 2013b (The MathWorks, Inc., Natick, MA) and the DIPY 0.13.0 library (http://nipy.org/dipy) under Python 3.6.4 (scipy 1.0.0) was used for MAP-MRI and MAPL. AMURA is implemented in MATLAB without multi-threading to report the results in Table 8.

Though raw execution times are an ambiguous performance index (they can be dramatically improved, for example, via GPU acceleration), they give a reasonable idea of the complexity of each method. Note the reported times for most of the methods make them unfeasible to be used on practical studies. In the case of RBF, they range from 5 to 24 days per volume, something that goes beyond the capability of clinical groups. Even in the best of the cases, MAPL is 17 times slower than AMURA. In all the cases, most of the time is spent in calculating the EAP. In MAPL, for instance, only 0.6% of the calculation time corresponds to the

**Table 8. Estimated execution times for the calculation of the measures with different methods.** One single volume (HCP MGH 1016) is processed.

| | Execution times | | |
|---|---|---|---|
| **Method** | **Two shells** | **Three shells** | **Four shells** |
| RBFs | 118h 10min | 332h 40min | 577h 12min |
| MAP-MRI | 13h 43min | 13h 46min | 16h 20min |
| MAPL | 2h 11min | 2h 14min | 2h 22min |
| AMURA | **6min 41s** | **7min 17s** | **8min 28s** |

measures, while the remaining 99.4% is spent in estimating the EAP. In the case of AMURA, 50% of the execution time corresponds to RTAP, since the estimation of the ODF is the most expensive operation, followed by RTPP, which takes 40% of the time. RTOP is the fastest measure, since it takes only 16s, 29s, and 54s for the different shells.

## Discussion and conclusions

AMURA are not intend to approximate the exact same numeric parameters as multi-shell methods compute. On the contrary, their aim is inferring micro-structural information related to, and with comparable discrimination power as, that revealed by MAP-MRI, MAPL, or RBF. Fig 1 and Table 3 evidence the anatomical consistency of AMURA, both visually and numerically. Tables 4, 5, and 6 confirm they are able to discriminate tissue properties in a similar way as multi-shell methods do.

With regard to the first issue, i.e. anatomical consistency, EAP-based measures explicitly account for the radial behavior of the diffusion signal, which is actually sampled. With AMURA, the radial behavior is not sampled but modeled as a mono-exponential decay. The hypothesis leading to the computation of the whole EAP should be, therefore, that the study of the whole EAP provides more specific/sensitive measures, i.e., there is certain anatomical information encoded in the radial behavior of the EAP that would remain hidden with AMURA. However, Table 3 highlights this is not always the case: different EAP-methods bring in less consistent results among them than some of them exhibit with AMURA for analogous measures (RTOP, RTPP, or RTAP). Paradoxically, the similarity between RBF and MAP-like methods even worsens as new shells with higher b-values are introduced.

As a first attempt to explain this behavior, we may recall that the measures computed are merely scalars, i.e., the complex information gathered in the whole 3-D domain of the EAP is somehow collapsed to one single number: the RTOP, for instance, is the value of the EAP at a single point (zero), which corresponds to the integration of the diffusion signal in the whole q-space, in a way that most of the information is lost in the average.

However, the averaging process behind the scalar measures does not explain why the corresponding outputs obtained from the different EAP-based methods do not converge to analogous values, or why the model-constrained AMURA measures seem to mimic MAPL values better than model-free, EAP-based MAP-MRI and RBF in Table 3. Moreover, as the number of shells and the number of samples per shell increase, MAPL, MAP-MRI, and RBF would be expected to converge to exactly the same values, since all of them estimate the same mathematical entity (the EAP) and all of them use the same mathematical description of the related measures (RTOP, RTPP, and RTAP). The experiments here reported show this is not always the case and, surprisingly, MAP-MRI and RBF tend to diverge from MAPL more than AMURA does.

Obviously, the mono-exponential model introduces a non-negligible error in the estimated measures. But the estimation of the EAP is by no means free of certain issues that compromise its accuracy: first, the EAP is usually represented as a superposition of functions selected from a basis or frame where the EAP is assumed to be sparse, which is only a rough approximation; second, the estimation is usually grounded on non-linear, iterative procedures, whose numerical stability is not always guaranteed and whose actual convergence is often conditioned by computational time restrictions; third, EAP estimation requires probing very strong diffusion gradients that drastically worsen the SNR, which may have an uncertain impact depending on the optimization method to be used; fourth, an additional side effect of the use of strong diffusion gradients is that the linear Fourier transform relation between the EAP and the diffusion signal, which is the keystone of all EAP-based methods, may no longer hold with accuracy due

to non-linearity, diffraction, and/or non-negligible diffusion during the application of pulsed gradients in a time $\delta$ (see Fig 8, where including the fourth shell in the estimation heavily increases all measures for MAPL; this might suggest the Fourier model has been compromised at this point).

The combination of these four factors (and possibly others) may affect each EAP-based method in very different ways, and they could even represent a larger error than that introduced by the mono-exponential model. This could possibly explain the discrepancies between the measures computed with any of the three EAP-based methods, especially the higher deviations of RBF when 3 shells (instead of 2) are used in Table 3. Of course, AMURA does not get rid of this issue. But, once again, the goal of AMURA is not estimating the exact same values as EAP-based methods: a shifted (level/contrast changed) version of a given measure will have exactly the same discriminant power as its former version, and therefore it will be equally valuable. Going back to Fig 8, EAP-based measures do not always respect this principle: RBF, for example, assigns very different, non-consistent relative values of RTAP among anatomies with similar FA depending on the number of shells used. Since RTAP is somehow related to the anisotropy (to the FA), this is by no means the expected behavior. MAP-like estimators, as well as AMURA, get rid of this artifact for RTAP but not for RTPP. However, since RTPP is not as closely related to the anisotropy as RTAP, and as long as AMURA-RTPP is still consistent with MAP-RTPP, this seems acceptable.

All in all, the *apparent* nature of AMURA makes corresponding measures heavily dependent on the measured shell (see Fig 6), but a similar variability is also found in multi-shell methods (Fig 8).

Once the consistency of AMURA has been thoroughly discussed, the big deal is their power to resolve micro-structural features beyond the capabilities of conventional DT-MRI. Tables 4, 5, and 6 suggest that AMURA might be as good as the other multi-shell techniques to distinguish different populations based on tissue properties. Going back to the previous discussion, the lack of consistency between the raw values of RTOP, RTPP, and RTAP computed with different multi-shell methods translates in similar discrepancies in the classification of white matter voxels. If AMURA correlated with MAPL stronger than the other multi-shell techniques did, they indeed provide better overlapped classifications too. Remarkably, AMURA finds two populations that more closely resemble those found by MAPL than MAP-MRI does, even when MAPL and MAP-MRI share a good number of common features. This remains true for all *apparent* measures at all available b-values. Hence, if we admit that EAP imaging provides measures that actually relate to micro-structural properties [12, 22, 29, 31], corresponding AMURA indexes should be assumed to probe actual tissue information as well. Once again, this claim can be justified only under the hypothesis that the radial integration to compute scalar measures blurs out a major part of the radial information within the q-space.

The experiment in Fig 4 supports this claim, at least for RTPP: while tensor-derived measures are not able to distinguish different populations within the corpus callosum, AMURA-RTPP finds two distinct regions that can be easily identified in Fig 5. In other words, AMURA-RTPP is measuring a micro-structural information that is not revealed with standard DT-MRI. Paying attention to Fig 5, the two populations distinguished by AMURA-RTPP become evident: in the outermost region, the corpus callosum is interleaved with the cingulum, so that restricted diffusion prevails, the maximum diffusivity decreases, and the RTPP increases (lobes at the right of the valley in Fig 4, rightmost column). In the innermost part, on the contrary, the corpus callosum is closer to the CSF and non-restricted diffusion takes a more relevant role: the maximum diffusivity increases and, as a consequence, the RTPP decreases (lobes at the left of the valley in Fig 4). At the sight of Fig 4, MAPL seems to find only subtle differences between these two populations, *performing worse* than AMURA.

Nonetheless, the PDD database comprises only 2 shells, and hence it is not particularly well suited for this technique.

In any case, RTPP yields statistically significant differences between controls and patients at both the GCC and the SCC in all cases, see Table 7 (though the AMURA-RTPP yields a higher significance). This is not the case for RTOP and RTAP. It is important to stress here that the aim of the experiment is not demonstrating the clinical usefulness of AMURA in the particular case of PD, but testing its capability to describe micro-structural features. In other words, the fact that RTOP and RTAP are not able to find significant differences between controls and PD patients only means that the micro-structural properties they describe do not seem to be altered by this particular pathology and/or in this particular data set.

One further step in the present study would be the validation of AMURA as clinical bio-marker candidates for diverse pathologies. Though Table 7 somehow points in this direction, this aspect must be thoroughly tested. In this sense, one major advantage of AMURA is its compatibility with nowadays standard acquisition protocols, so that they can be computed over already existing data sets such as the PDD database. Indeed, in case several shells with different b-values are available in one such database (as it is the case with PDD), AMURA can be trivially extended to fit the mono-exponential model to the entire data set and obtain more robust markers. On the contrary, multi-shell methods like MAPL need *ad-hoc* new acquisitions to attain satisfactory results, which complicates their clinical validation.

Moreover, since AMURA avoids the estimation of the actual EAP, the computation of its related measures may be done in a fast and robust way, i.e., without imposing a computational burden to the standard protocols: some of the experiments in the present paper report an acceleration about three orders of magnitude ($10^3$) compared to EAP-based measures, see Table 8. A whole volume can be processed in 6 to 8 minutes, so that a clinical study with 200 different subjects could be finished in 26 hours. The same cohort would take 4808 days (RBF), 135 days (MAP-MRI), or 19 days (MAPL), which obviously limits the applicability of these methods. The computational simplicity of AMURA, however, does not only imply faster execution times, but also more robust estimations due to its closed-form. As opposed, EAP-based techniques usually estimate the whole EAP from multi-shell samplings based on iterative procedures, which, as discussed above, lead to high discrepancies in the output measures.

On the other side of the coin, the major drawback behind AMURA is the explicit assumption of a specific radial behavior for the diffusion, which cannot fit the whole q-space. As a consequence, the selection of a particular b-value may change the anatomical measures that have been consequently dubbed *apparent*. However, as we have shown, this dependence on the b-value can also be found in other state of the art methods (see Fig 8), whose outputs vary with the number of shells used for the estimation of the EAP. This implies the results of clinical trials could be compared against each other only if the same b-value is preserved across the studies. This is by no means something new to diffusion imaging: it is well-known that a change in the acquisition parameters (number of gradients, b-value, resolution, scanner vendor, etcetera) seriously affects scalar measures like the FA or the MD [47, 48].

## Appendix: Calculation of the structural measures using the diffusion tensor

If a Gaussian diffusion propagator is assumed, $P(\mathbf{R})$ is a mixture of independent and (nearly) identically distributed bounded cylinder statistics and, by virtue of the central limit theorem, their superposition is Gaussian distributed. The measured signal in the q–space is the (inverse)

Fourier transform of the PDF and it can be expressed as:

$$E(\mathbf{q}) = \mathfrak{F}^{-1}\{P(\mathbf{R})\}(\mathbf{q}) = \exp(-4\pi^2\tau\mathbf{q}^T\mathcal{D}\mathbf{q}), \tag{15}$$

which represents the well-known Stejskal–Tanner equation [49]. The diffusion tensor $\mathcal{D}$ is the anisotropic covariance matrix of the Gaussian PDF $P(\mathbf{R})$, and therefore it is a symmetric, positive–definite matrix with positive eigenvalues and orthonormal eigenvectors. If we use this model to estimate the measures, we obtain:

$$\begin{aligned} \text{RTOP} &= \frac{1}{\sqrt{(4\pi\tau)^3}} \cdot (\lambda_1 \cdot \lambda_2 \cdot \lambda_3)^{-1/2} \\ &= \text{RTPP} \cdot \text{RTAP}; \end{aligned} \tag{16}$$

$$\text{RTPP} = \frac{1}{\sqrt{4\pi\tau}} \cdot (\lambda_1)^{-1/2}; \tag{17}$$

$$\text{RTAP} = \frac{1}{\sqrt{(4\pi\tau)^2}} \cdot (\lambda_2 \cdot \lambda_3)^{-1/2}, \tag{18}$$

where $\lambda_1 \geq \lambda_2 \geq \lambda_3$ are the three real, non-negative eigenvalues of $\mathcal{D}$.

## Acknowledgments

This work was supported by Ministerio de Ciencia e Innovación of Spain with research grants RTI2018-094569-B-I00 and PRX18/00253 (Estancias de profesores e investigadores senior en centros extranjeros). Maryam Afzali is supported by a Wellcome Trust grant (096646/Z/11/Z). Tomasz Pieciak acknowledges National Science Centre (Poland) for funding resource (2015/19/N/ST7/01204).

The authors acknowledge Lipeng Ning, Carl-Fredrik Westin and Yogesh Rathi from Brigham and Women's Hospital, Harvard Medical School for sharing the source code of directional RBFs and their outstanding assistance. The authors thank the contributors of DIPY project (http://nipy.org/dipy/) for providing the MAP-MRI basis implementation.

Data collection and sharing for this project was provided by (1) the *Human Connectome Project* (HCP; Principal Investigators: Bruce Rosen, M.D., Ph.D., Arthur W. Toga, Ph.D., Van J. Weeden, MD). HCP funding was provided by the National Institute of Dental and Craniofacial Research (NIDCR), the National Institute of Mental Health (NIMH), and the National Institute of Neurological Disorders and Stroke (NINDS). HCP data are disseminated by the Laboratory of Neuro Imaging at the University of Southern California; (2) the *High-quality diffusion-weighted imaging of Parkinson's disease* data base, Cyclotron Research Centre, University of Liège.

## Author Contributions

**Conceptualization:** Santiago Aja-Fernández, Antonio Tristán-Vega.

**Data curation:** Malwina Molendowska, Tomasz Pieciak, Antonio Tristán-Vega.

**Formal analysis:** Santiago Aja-Fernández, Antonio Tristán-Vega.

**Investigation:** Santiago Aja-Fernández, Antonio Tristán-Vega.

**Methodology:** Santiago Aja-Fernández, Antonio Tristán-Vega.

**Project administration:** Santiago Aja-Fernández.

**Resources:** Maryam Afzali.

**Software:** Santiago Aja-Fernández, Maryam Afzali, Antonio Tristán-Vega.

**Supervision:** Santiago Aja-Fernández, Rodrigo de Luis-García, Antonio Tristán-Vega.

**Validation:** Santiago Aja-Fernández, Malwina Molendowska, Tomasz Pieciak, Antonio Tristán-Vega.

**Visualization:** Santiago Aja-Fernández, Antonio Tristán-Vega.

**Writing – original draft:** Santiago Aja-Fernández, Antonio Tristán-Vega.

**Writing – review & editing:** Santiago Aja-Fernández, Antonio Tristán-Vega.

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
