## [Decision Letter · Decision Letter 0]

21 Nov 2019

PONE-D-19-27308

Micro-structure diffusion scalar measures from reduced MRI acquisitions

PLOS ONE

Dear Dr. Tristán-Vega,

Thank you for submitting your manuscript to PLOS ONE. After careful consideration, we feel that it has merit but does not fully meet PLOS ONE’s publication criteria as it currently stands. Therefore, we invite you to submit a revised version of the manuscript that addresses the points raised during the review process.

We would appreciate receiving your revised manuscript by Jan 05 2020 11:59PM. To enhance the reproducibility of your results, we recommend that if applicable you deposit your laboratory protocols in protocols.io, where a protocol can be assigned its own identifier (DOI) such that it can be cited independently in the future. For instructions see: http://journals.plos.org/plosone/s/submission-guidelines#loc-laboratory-protocols

We look forward to receiving your revised manuscript.

Kind regards,

Pew-Thian Yap

Academic Editor

PLOS ONE

Journal Requirements:

2. We note that Figure(s) [#] in your submission contain copyrighted images. All PLOS content is published under the Creative Commons Attribution License (CC BY 4.0), which means that the manuscript, images, and Supporting Information files will be freely available online, and any third party is permitted to access, download, copy, distribute, and use these materials in any way, even commercially, with proper attribution. For more information, see our copyright guidelines: http://journals.plos.org/plosone/s/licenses-and-copyright.

1.         You may seek permission from the original copyright holder of Figure(s) [#] to publish the content specifically under the CC BY 4.0 license.

3. We note that your paper includes detailed descriptions of individual patients/participants. As per the PLOS ONE policy (http://journals.plos.org/plosone/s/submission-guidelines#loc-human-subjects-research) on papers that include identifying, or potentially identifying, information, the individual(s) or parent(s)/guardian(s) must be informed of the terms of the PLOS open-access (CC-BY) license and provide specific permission for publication of these details under the terms of this license. Please download the Consent Form for Publication in a PLOS Journal (http://journals.plos.org/plosone/s/file?id=8ce6/plos-consent-form-english.pdf). The signed consent form should not be submitted with the manuscript, but should be securely filed in the individual's case notes. Please amend the methods section and ethics statement of the manuscript to explicitly state that the patient/participant has provided consent for publication: “The individual in this manuscript has given written informed consent (as outlined in PLOS consent form) to publish these case details”.

Reviewers' comments:

Reviewer's Responses to Questions

**Comments to the Author**

1. Is the manuscript technically sound, and do the data support the conclusions?

Reviewer #1: Yes

Reviewer #2: Yes

2. Has the statistical analysis been performed appropriately and rigorously? 

Reviewer #1: Yes

Reviewer #2: Yes

3. Have the authors made all data underlying the findings in their manuscript fully available?

Reviewer #1: Yes

Reviewer #2: Yes

4. Is the manuscript presented in an intelligible fashion and written in standard English?

Reviewer #1: Yes

Reviewer #2: Yes

5. Review Comments to the Author

Reviewer #1: The purpose of this work is to present a method for the estimation of the microstructural scale, e.g. RTOP, RTPP, and RTAP from single-shell acquisitions. The paper presents a simplified formulation to calculate these scales with the assumption that the diffusion anisotropy is roughly independent of the radial direction. The proposed work proves its usability in the different acquisitions. The manuscript is concise and well explained and organized. I have a few comments on the manuscript:

1. "...assuming the diffusion anisotropy is roughly independent of the radial direction.", Please have an explaining of the rationality of this hypothesis for clinical applications. Also, what are the special requirements for the acquisition scheme?

2. Please also give some results to explain the impact of the number of gradient samples (or angular resolution) on the results.

3. Note that the experiment implemented in higher b-value on the result, e.g. Figure.1 and Table.3. What happened with b=1000?

4. Please clarify the definition of the correlation coefficient in Table.3.

5. In Figure.1, the RTPP calculated by AMURA seems to be significantly different from the results from other others. which is obviously cleaner. It seems to be related to the hypothesis that diffusion anisotropy is roughly independent of the radial direction. But it's still unclear whether this kind of change is an improvement. So it is better to show the correlation of RTPP with MD or RD.

Reviewer #2: In this paper, the authors propose a set of novel diffusion indices, called apparent measures using reduced acquisitions (AMURA), which can mimic the EPA-derived indices, but require only single-shell dMRI data. AMURA are based on the observation that EAP-derived indices are radial integrations of a set of measures at different shells. Based on this observation, the authors simplify the acquisition to a single-shell case and propose a set of single-shell measures to mimic EPA-derived indices, including RTOP, RTPP, and RTAP. The authors first show the analytical solutions for AMURA and then provide detailed numerical implementations by using spherical harmonics. AMURA is evaluated using a normal dataset, HCP, and a patient dataset, PPD. Three baseline methods, including RBF, MAPL, and MAP-MRI, are involved in the evaluation. Extensive experimental results demonstrate the effectiveness of AMURA, both qualitatively and quantitatively.

The paper is written well and motivation is clearly stated. AMURA are novel indices, which allow the study of brain disorders using EPA-like indices and clinical dMRI data. Regarding the proposed method, I have several minor concerns.

1. AMURA change at different b-values. Is there an optimal b-value for AMURA? For instance, which b-value gives the best AMURA that are most similar to the EAP-derived indices?

2. Page 10: A threshold is selected in the valley right after the main lobe in each case (for RTOP: 2 · 10^6 mm^−3 ; for RTPP: 90 mm^−1; for RTAP: 1.5 · 10^4 mm^−2 ). Please clarify whether all comparison methods share the same set of thresholds, or not.

3. The experimental results are convincing and sufficiently demonstrate the effectiveness of AMURA. However, I am wondering whether synthetic data experiments are useful or not since we do not have ground truth EAP measures in real data experiments.

6. PLOS authors have the option to publish the peer review history of their article (what does this mean?). If published, this will include your full peer review and any attached files.

Reviewer #1: Yes: Ye Wu

Reviewer #2: No

---

## [Author Response · Author response to Decision Letter 0]

5 Jan 2020

Please, see the "answers to reviewers" letter attached.

---

## [Decision Letter · Decision Letter 1]

10 Feb 2020

Micro-structure diffusion scalar measures from reduced MRI acquisitions

PONE-D-19-27308R1

Dear Dr. Tristán-Vega,

We are pleased to inform you that your manuscript has been judged scientifically suitable for publication and will be formally accepted for publication once it complies with all outstanding technical requirements.

With kind regards,

Pew-Thian Yap

Academic Editor

PLOS ONE

Additional Editor Comments (optional):

Reviewers' comments:

Reviewer's Responses to Questions

**Comments to the Author**

1. If the authors have adequately addressed your comments raised in a previous round of review and you feel that this manuscript is now acceptable for publication, you may indicate that here to bypass the “Comments to the Author” section, enter your conflict of interest statement in the “Confidential to Editor” section, and submit your "Accept" recommendation.

Reviewer #1: All comments have been addressed

Reviewer #2: All comments have been addressed

2. Is the manuscript technically sound, and do the data support the conclusions?

Reviewer #1: Yes

Reviewer #2: Yes

3. Has the statistical analysis been performed appropriately and rigorously? 

Reviewer #1: Yes

Reviewer #2: Yes

4. Have the authors made all data underlying the findings in their manuscript fully available?

Reviewer #1: Yes

Reviewer #2: Yes

5. Is the manuscript presented in an intelligible fashion and written in standard English?

Reviewer #1: Yes

Reviewer #2: Yes

6. Review Comments to the Author

Reviewer #1: The authors have properly resolved all of my concerns and substantially improved the manuscript. So I would recommend its publication.

Reviewer #2: All comments have been well addressed in this revised version. I do not have any additional comments. This paper can be accepted.

7. PLOS authors have the option to publish the peer review history of their article (what does this mean?). If published, this will include your full peer review and any attached files.

Reviewer #1: Yes: Ye Wu

Reviewer #2: No

---

## [Editor Report · Acceptance letter]

13 Feb 2020

PONE-D-19-27308R1 

Micro-structure diffusion scalar measures from reduced MRI acquisitions 

Dear Dr. Tristán-Vega:

I am pleased to inform you that your manuscript has been deemed suitable for publication in PLOS ONE. Congratulations! Your manuscript is now with our production department. 

With kind regards,

on behalf of

Dr. Pew-Thian Yap 

Academic Editor

PLOS ONE